# Synthesis of Coal-Fly-Ash-Based Ordered Mesoporous Materials and Their Adsorption Application

**DOI:** 10.3390/ma16072868

**Published:** 2023-04-04

**Authors:** Miaomiao Tan, Dahai Pan, Shuwei Chen, Xiaoliang Yan, Lina Han, Ruifeng Li, Jiancheng Wang

**Affiliations:** 1College of Chemical Engineering and Technology, Taiyuan University of Technology, Taiyuan 030024, China; 2College of Chemistry, Taiyuan University of Technology, Taiyuan 030024, China; 3College of Materials Science and Engineering, Taiyuan University of Technology, Taiyuan 030024, China; 4State Key Laboratory of Clean and Efficient Coal Utilization, Taiyuan University of Technology, Taiyuan 030024, China; wangjiancheng@tyut.edu.cn

**Keywords:** coal fly ash, ordered mesoporous materials, subcritical activation treatment, Al doping, adsorption

## Abstract

A feasible approach was developed for the synthesis of ordered mesoporous SBA-15-type materials using coal fly ash (CFA) as raw material. In the proposed approach, CFA was, firstly, activated by subcritical water with the addition of NaOH, which allowed an efficient extraction of silicon species from CFA under strong acidic conditions at near room temperature. Subsequently, in the synthesis system, using silicon extraction solution as the silicon precursor, the introduction of anhydrous ethanol as a co-solvent effectively inhibited the polymerization of silanol species and promoted their collaborative self-assembly with surfactant molecules by enhancing the hydrogen bond interactions. The resultant SBA-15 material had a high purity, high specific surface area (1014 m^2^/g) and pore volume (1.08 cm^3^/g), and a highly ordered mesostructure, and, therefore, exhibited an excellent removal efficiency (90.5%) and adsorption capacity (160.8 mg/g) for methylene blue (MB) from simulated wastewater. Additionally, the generation of surface acid sites from the homogenous incorporation of Al atoms onto the mesoporous walls of SBA-15 combined with the perfect retention of the ordered mesostructure endowed the obtained Al-SBA-15 material with a further boost in the removal performance of MB. The MB removal efficiency can reach ~100%, along with a maximum adsorption capacity of 190.1 mg/g.

## 1. Introduction

Coal fly ash (CFA) generated from the combustion of coal in electric power plants is one of the main solid wastes in China, whose emissions are increasing exponentially due to the rapid demand for energy [1]. The efficient recycling of CFA has been a key issue, due to the accumulation, disposal, and under-utilization of massive amounts of CFA, posing serious hazards to the environment and human health [2]. Taking CFA containing high amounts of silica (40−60 wt.%) into consideration, more attention is focused on the synthesis of nanoporous materials using CFA as an extensive, cheap, and readily available silicon source, which provides an available option for recycling CFA to prepare high-value-added products.

With the fast development of industry, the mass discharge of pollutants triggers an increasing water contamination, which exerts a drastic threat to the ecological system. Taking the common organic dye methylene blue (MB) for example, its release into water sources can reduce light penetration, and, in turn, affects photosynthesis and induces damage to the ecosystem’s balance [3]. In addition, prolonged exposure to MB may provoke many diseases, such as eye and skin irritation, digestive infections, and mental confusion, for both aquatic animals and humans [4]. The key issue in treating contaminated water is the development of high-performance adsorbents, since the adsorption technology is characterized by high efficiency, simple operation, and environmental friendliness, as compared with other wastewater treatment techniques (such as chemical precipitation, membrane filtration, degradation, and so on) [5]. Ordered mesoporous silica materials with uniform and adjustable channels, high specific surface areas and pore volumes, and large and tunable pore sizes provide important impetuses for the development of high-performance adsorbents [6] and are widely used for the efficient removal of heavy metals and toxic organic dyes [7,8,9]. Among various ordered mesoporous silica materials, SBA-15 type materials have received much attention, owing to their non-toxicity and high structural stability [10]. Unfortunately, when CFA was used as a raw material, typical recipes for the preparation of SBA-15 did not act effectively, which could be attributed to the low extraction efficiency of silicon species from CFA [11]. Consequently, costly silicon precursors (such as TEOS) should be added to improve the quantity of the product [12,13]. In addition, the strong acidic synthesis conditions make it is extremely difficult to directly incorporate metal atoms into the siliceous framework of SBA-15 because metals only exist in their aqueous cationic form rather than their corresponding oxo species [14]. As a result, SBA-15 with a neutral Si framework has a weak acidity and lacks active sites, which severely hinder its practical applications [15]. Therefore, it is still a great challenge to develop more economical procedures for the synthesis of metal-doping SBA-15 materials with excellent structural, textural, and surface properties using CFA as raw material.

Recently, we introduced a novel supercritical water treatment method to activate coal gangue for extracting Si species [16]. Compared to the extensively used high-temperature alkali fusion method (a high temperature (600–900 °C) and a high usage amount of alkali were required), our method involves a much lower activation temperature, a shorter activation time, and a smaller usage amount of alkali; thus, it can effectively reduce the energy consumption. More importantly, through our method, an Si extraction efficiency of 69.2% from the coal gangue can be obtained, which is much higher than the reported activation methods such as high-temperature alkali fusion [17], strong alkali dissolution [18], mechanical activation [19], and so on. Inspired from our previous work [16], herein, CFA was activated using the subcritical water with the addition of NaOH, and ordered mesoporous siliceous SBA-15 material, derived from CFA, was designed and successfully synthesized. In addition, to improve the surface acidity and adsorption performance of the obtained SBA-15, aluminum (Al) atoms were successfully incorporated into the silica framework by adjusting the pH hydrothermal grafting method. The resultant materials of SBA-15 and Al-SBA-15 possessed highly ordered two-dimensional (2D) hexagonal mesostructures, narrow mesoporous size distributions, and high specific surface areas and pore volumes, and they were evaluated as efficient adsorbents for the removal of methylene blue (MB) from the simulated wastewater. The subcritical activation mechanism of CFA, the crucial role of ethanol as a co-solvent in promoting the collaborative self-assembly between the inorganic silicon species and the organic surfactant molecules, and the promoting effect of Al doping on MB removal performance were investigated.

## 2. Materials and Methods

### 2.1. Raw Materials and Chemicals

Coal fly ash (CFA) was derived from a coal-fired power plant (Shanxi, China). Triblock copolymer P123 (EO_20_PO_70_EO_20_) was obtained from Sigma-Aldrich Pty., Ltd., St. Louis, MI, USA. Sodium hydroxide (NaOH), hydrochloric acid (HCl), acetic acid (HAc), aluminum nitrate nonahydrate (Al(NO_3_)_3_·9H_2_O), and anhydrous ethanol were purchased from Sinopharm Group Chemical Reagent Co., Ltd., China. All commercially available chemical reagents were of an analytical grade and used without further purification.

### 2.2. Preparation of SBA-15-Type Mesoporous Materials

As shown in Figure 1, the synthesis of ordered mesoporous Al-SBA-15 was carried out in three stages: (1) silicon extraction from CFA, (2) preparation of siliceous SBA-15, and (3) grafting Al atoms into the mesoporous silica framework of SBA-15.

Silicon extraction was performed through a subcritical hydrothermal method. The as-received CFA (10 g) was mixed with 120 mL of NaOH solution (0.5 mol/L) in an autoclave reactor, and the mixture was reacted at 250 °C with a pressure of 400 kPa for 2 h under stirring (500 rpm). The activated CFA was collected by filtration, washed, dried in air, and dispersed in 140 mL of 2.5 M HCl solution. After vigorously stirring at 45 ºC for 1.5 h, the supernatant silicon-rich solution was obtained using a filtration and separation process, which was used as silicon precursor for the synthesis of mesoporous silica materials. 

During the preparation of the as-synthesized siliceous SBA-15, 2.2 g of P123 and 5.5 g of HAc were dissolved in 18 mL of anhydrous ethanol, followed by the slow addition of silicon extraction solution prepared in the previous step under vigorous stirring at 38 °C. After reacting for 24 h, the mixture was transferred into an autoclave for solvothermal treatment at 100 °C for 24 h. The resulting precipitate was filtered and washed with deionized water.

During the third stage, the ordered mesoporous Al-SBA-15 was prepared using a post-synthesis grafting method. The as-synthesized SBA-15 was mixed with 60 g of deionized water containing 0.677 g of Al(NO_3_)_3_·9H_2_O. The pH value of the solution was adjusted to 1.65 using 0.5 M HCl. The mixture was transferred into an autoclave for hydrothermal Al-grafting treatment at 100 °C for 24 h. The final product was collected by filtration, washed, dried, calcined at 550 °C for 5 h to remove the organic template, and named SA-E. For comparison, siliceous SBA-15 material prepared during the second stage with a calcination treatment at 550 °C for 5 h was denoted as S-E. 

In order to investigate the effect of co-solvent anhydrous ethanol on the synthesis of siliceous SBA-15, 18 mL of deionized water was used instead of anhydrous ethanol for the dissolution of template P123. The mesoporous silica material was synthesized following the same procedure as that for preparing S-E, and the final product was designated as S-H. In addition, a SBA-15 material was also prepared by the conventional method using the chemical TEOS as silicon precursor [20] and named S-O.

### 2.3. Adsorption Procedure

Adsorption of methylene blue (MB) dye was conducted using batch balance experiments. At 25 °C, a required amount (0.1, 0.3, and 0.5 g/L) of obtained mesoporous material (including S-O, S-H, S-E, and SA-E) was suspended in a 50 mL of 10–70 mg/L MB solution with various pH values under stirring at 200 rpm. The sampling was performed at different time intervals. The MB content in the solution before and after the adsorption was determined using a UV-vis spectrophotometer at λ = 664 nm. The removal efficiency (*Y*) and adsorption capacity (*Q_e_*) were calculated using Equations (1) and (2), respectively.
(1)$$Y=\frac{C_{0}-C_{e}}{C_{0}}\times 100$$
(2)$$Q_{e}=\frac{\left(-C_{e}\right)V}{m}\;$$
where *C*_0_ and *C*_e_ stand for the initial and equilibrium concentration of MB in solution, respectively (mg/L), *V* denotes the volume of MB solution (L), and *m* is the adsorbent weight (g).

### 2.4. Characterization

The chemical compositions of both the as-received CFA and subcritical activated CFA were analyzed using a Panalytical Epsilon1 X-ray fluorescence (XRF) spectrometer. The concentration of the silicon-rich solution was measured using a Thermo-Fisher ICAP 6300 inductively coupled plasma optical emission spectrometer (ICP-OES). Powder X-ray diffraction (XRD) patterns were recorded on a Rigaku MiniFlex 600 diffractometer using Ni-filtered Cu Kα radiation. Transmission electron microscopy (TEM) was carried out on a JEM 2100 microscope operated at 200 kV. Nitrogen adsorption–desorption isotherms were measured on an ASAP 2460 analyzer at 77 K. The BET method was used for the calculation of specific surface areas, while the BJH method was used for the determination of pore size distributions derived from the adsorption branches of isotherms. The total pore volumes were calculated according to the amount of nitrogen adsorbed at the relative pressure of 0.99. The in situ Fourier transform infrared (FTIR) spectra were obtained using a Bruker TENSOR II FTIR spectrometer. Before analysis, the samples were pressed into wafers with the same weight and area and then mounted onto a high-vacuum cell for dehydration treatment under a high vacuum (10^−6^ Pa) at 140 °C for 5 h. After the complete removal of physically adsorbed water, the concentration of silanol (Si-OH) groups existing within the mesoporous silica wall of samples was determined according to the integral area of the Si-OH absorbent band. The size of silanol species existing in the acid-leaching desilication solution with and without the introduction of anhydrous ethanol were measured on a Microtrac MN42X laser diffraction particle size analyzer. The thermogravimetric analysis (TGA) was performed using a NETZSCH STA 449F3 instrument. The samples were heated in air with the heating rate of 10 °C/min from 100 °C to 500 °C. Elemental mapping measurements were performed on a JEOL JSM-7900F scanning electron microscope instrument with an energy dispersive X-ray (EDX) spectrometer. The acidity of samples was measured by the temperature-programmed desorption of NH_3_ (NH_3_-TPD) using a Micromeritics AUTOCHEM II 2920 analyzer that was equipped with a thermal conductivity detector (TCD). Before measurement, 100 mg of the sample was pretreated at 550 °C for 1 h and then cooled to 100 °C in a pure He flow. A 10% NH_3_-He flow (25 mL/min) was used to saturate the sample with ammonia for 30 min and then replaced with a pure He flow (25 mL/min) to remove the physically adsorbed NH_3_ for 1 h. Subsequently, the NH_3_-TPD was carried out from 100 °C to 550 °C at a rate of 10 °C/min in a pure He flow. The desorbed NH_3_ was monitored using TCD. The adsorption of pyridine was conducted using a TENSOR II FTIR instrument, and the types of acid sites in the samples were evaluated. Before the adsorption of pure pyridine vapors at room temperature, the sample was pretreated at 500 °C for 4 h under a high vacuum to remove the impurities adsorbed in the sample. The FTIR spectra of the sample, evacuated at 150, 250 and 350 °C, were recorded using a Bruker TENSOR II FTIR spectrometer. The concentration of MB was determined using a Shanghai MC-721 spectrophotometer.

## 3. Results and Discussion

### 3.1. The Subcritical Activation of CFA

The chemical compositions of as-received and activated CFA and the elemental concentrations in the acid leaching desilication solution (silicon-rich solution) derived from the activated CFA are listed in Table 1.

The principal components of the original CFA were SiO_2_ (47.9 wt.%), Al_2_O_3_ (36.3 wt.%), and Fe_2_O_3_ (6.2 wt.%). After high-temperature subcritical water treatment, an evident increase in the content of SiO_2_ (52.9 wt.%) was observed for the activated CFA. To illustrate the Si activation mechanism of CFA, the mineral composition of CFA before and after the activation treatment was characterized using XRD analysis. As shown in Figure 2, the major crystalline components of the original CFA were stable mullite and quartz along with other amorphous aluminosilicates. For activated CFA, the diffraction peaks of mullite and quartz disappeared; meanwhile, a new crystalline phase, square zeolite (Analcime, Na(Si_2_Al)O_6_·H_2_O; JCDPS 41-1478) [21], was detected through the diffraction peaks at the 2θ values of 15.76°, 18.22°, 25.92°, 30.50°, 33.20°, 35.78°, 40.46°, 47.72°, and 52.46°, which were assigned to the (211), (220), (400), (332), (431), (521), (611), (640), and (651) reflections of Analcime, respectively. This result indicates that the introduction of alkali can efficiently enhance the reactivity of SiO_2_ and Al_2_O_3_ and broke the aluminosilicates framework of mullite to free the elements of Si and Al during the high-temperature subcritical water treatment process [22]. Under the promotion of Na^+^, the secondary reaction between the dissolved SiO_3_^2−^ and Al(OH)_4_^−^ accounted for the formation of square zeolite [23] and the increased content of SiO_2_ for the activated CFA. The results from the elemental analysis confirmed that the square zeolite in activated CFA displayed a high solubility in the HCl solution. A silicon concentration of 0.258 mol/L was found in the acid leaching solution (Table 1). Based upon the calculations performed considering the amount of silica in the original CFA and the amount of silica actually dissolved in the extraction solution, the extraction efficiency of silica reached the value of 46.9%, which was relatively higher or comparable compared to the earlier reported Si extraction techniques (for example, alkali fusion [24], the ammonium fluoride microwave-assisted method [25], and the alkaline-acid leaching process [26]). Even so, it should be pointed out that the silicon concentration in the extraction solution was still far below the required concentration (about 0.33 mol/L) for the synthesis of SBA-15 material using the conventional synthesis method [27].

### 3.2. The Role of Ethanol as a Co-Solvent for the Synthesis of SBA-15-Type Materials from CFA

The supernatant acid leaching solution of silicon species extracted from CFA was used as the only silicon source to prepare mesoporous silica materials (Samples S-H and S-E). As a reference, the chemical TEOS was also used as the only silicon source for the preparation of ordered mesoporous SBA-15 material (Sample S-O). The small-angle XRD patterns of obtained samples of S-O, S-E, and S-H, prepared with different synthesis solutions, are shown in Figure 3a. Sample S-O, prepared using TEOS and water as silicon source and solvent, respectively, displayed three well-resolved diffraction peaks at 2ϴ = 0.97, 1.63, and 1.90°. According to the *d*-spacing reciprocal ratio, such three diffraction peaks can be indexed as the (100), (110), and (200) reflections of an ordered 2D hexagonal mesostructure (space group *p*6*mm*), respectively. Different from Sample S-O, Sample S-H, prepared using silicon species extracted from CFA as silicon precursor and water as the only solvent, only exhibited a strong peak at 0.90° in its XRD pattern. The absence of the other two weak diffraction peaks suggests that Sample S-H possesses a disordered worm-like mesoporous structure [28], corresponding to a less ordering in the mesoporous arrangement. This indicates that it is difficult to synthesize a highly ordered mesoporous SBA-15 material in an aqueous solution using silicon species extracted from CFA as a silicon source due to the relatively low silicon concentration (Table 1). Such a conclusion is in accordance with the previously reported results [29]. Interestingly, for the Sample S-E, prepared using the anhydrous ethanol as co-solvent, an intense diffraction peak with a narrower full width at half-maximum and two additional weak diffraction peaks were observed at 0.94°, 1.56°, and 1.78°, respectively. These three diffraction peaks could be assigned to the (100), (110), and (200) reflections, respectively [30]. Such a result revealed that the introduction of anhydrous ethanol in the synthesis solution was favorable for the synthesis of siliceous SBA-15 material with an ordered 2D hexagonal mesostructure (space group *p*6*mm*) when CFA was used as the raw material. According to the intense (100) diffraction peak, a *d*_100_ spacing of 9.39 nm was calculated for Sample S-E, which corresponded to a unit cell parameter (*a*) of 10.84 nm (Table 2). In addition, no diffraction peak was observed in the wide-angle XRD pattern of Sample S-E (Figure 3b), suggesting that Sample S-E was amorphous in nature. The chemical composition from ICP analysis showed that the content of Si in Sample S-E accounted for 100%. The absence of metal elements, such as Al, Fe, and Ti can be attributed to the use of strong acid synthesis solution, in which metals exist only in their cationic form and are unable to be introduced into the mesoporous siliceous skeleton of S-E.

TEM images of Sample S-E further provided a direct observation of highly ordered mesoporous arrangements. As seen in Figure 4a,b, honeycomb-like hexagonally packed mesopores and cylindrical parallel mesoporous channels along the (100) and (110) directions can be clearly discerned, respectively, indicating that Sample S-E has a typical 2D hexagonal ordered mesostructure with a uniform mesoporous size and wall thickness [31]. Based upon the dark and white contrast of TEM images, the distance between the two adjacent mesopores was ca. 10.52 nm, which is consistent with the value of the unit cell parameter calculated from the small-angle XRD analysis (Figure 3a). In addition, the results from SAED analysis (the inset in Figure 2b) further confirmed that Sample S-E is amorphous in the nature of mesoporous walls, which is well consistent with the observation from the wide-angle XRD analysis (Figure 3b).

To ascertain the effect of introducing anhydrous ethanol as the co-solvent on the textural properties of resultant Sample S-E, N_2_ physisorption measurement was performed. Figure 5 shows the N_2_ adsorption–desorption isotherms and the corresponding pore size distribution curves of Samples S-O, S-E and S-H, which were prepared with different synthesis solutions [32]. The detailed textural properties of samples are summarized in Table 2. From Figure 5a, it can be seen that all Samples S-O, S-E and S-H exhibited a typical Type IV isotherm with a Type H1 hysteresis loop, which is typical characteristics of mesoporous materials with large cylindrical mesopores. It is noteworthy that, compared with the Sample S-H, the Samples S-O and S-E demonstrated a steeper capillary condensation step occurring at a relative pressure (P/P_0_) within the range of 0.65–0.75, which corresponded to a narrower mesopore size distribution (Figure 5b). Additionally, the porosities reflected from the isotherms of Samples S-O and S-E were much higher than that of Sample S-H. Based upon the calculations, the specific surface areas, pore volumes, and average mesoporous sizes of Samples S-O, S-H, and S-E were 768, 855, and 1014 m^2^/g; 1.09, 0.90, and 1.08 cm^3^/g; and 7.84, 8.62, and 7.06 nm, respectively (Table 2). The highly regular and ordered mesostructure can be responsible for the significantly improved textural properties for Sample S-E, as compared with those of Sample S-H. In addition, as shown in Figure 5b, the Samples S-E and S-H demonstrated higher microporous content than that of Sample S-O, which can account for the higher specific surface areas for Samples S-E and S-H than that of Sample S-O.

According to the characterization results from XRD, TEM, and N_2_ physisorption analyses, it can be inferred that the introduction of anhydrous ethanol as a co-solvent played a crucial role in promoting the collaborative self-assembly between the surfactant P123 molecules and the silicon species extracted from CFA for the synthesis of ordered mesoporous SBA-15 material with excellent properties. To understand the role of introduced anhydrous ethanol, high-vacuum in situ FTIR analysis was used to semi-quantitatively compare the relative concentration of silanol existing within the mesoporous walls of Samples S-E and S-H [33]. The relative silanol concentration was estimated from the integral area of the Si-OH band after completely removing the physisorbed water (Figure 6). Taking Sample S-E as an example, as shown in Figure 6a, before the evacuation treatment under high vacuum, its FTIR spectrum showed one sharp adsorption peak at 1630 cm^−1^, which can be ascribed to the bending mode of water molecules [34]. The broad adsorption band at 2800–3900 cm^−1^ can be attributed to the stretching hydroxyl (*ν*O-H) from the contribution of both the silanol groups and the physisorbed water, provided no other impurity components were present in Sample S-E. After evacuation treatment under a high vacuum, the adsorption peak at 1630 cm^−1^ could not be detected in the FTIR spectrum of Sample S-E, confirming that the physisorbed water was completely removed. Therefore, the integral area of the *ν*O-H band can be used to estimate the relative concentration of silanol groups.

Figure 6b shows the FTIR spectra of Samples S-E and S-H after evacuation treatment. It can be seen that a larger integral area of the *ν*O-H band was distinguished for Sample S-E, compared to Sample S-H, indicating that the silanol concentration of Sample S-E was much higher than that of Sample S-H. Such an observation strongly suggests that the introduction of anhydrous ethanol with weak polarity as a co-solvent combined with the coordination role of HAc can effectively inhibit the polymerization of silanol species in the acid-leaching solution extracted from CFA. As a result, a large number of silanol species with enough small sizes remained in the synthesis solution, which is vital in enhancing the hydrogen bond interactions between the silicon species and the hydrophilic segment (PEO) of surfactant P123 [35], thus driving their collaborative self-assembly to form an organic–inorganic hybrid composite with an ordered mesoporous structure. On the contrary, when only water was used as the solvent, the rapid hydrolysis and condensation of silicon species catalyzed by the hydrochloric acid led to the generation of polymeric silicon species and/or nano-silica particles with low silicon content and a prominently increased size, due to which the hydrogen bond interactions between the silicon species and the surfactant molecules were significantly weakened. Taking this fact into account, once the CFA extraction solution with a very low silicon concentration was used as the only silicon source, it was difficult to prepare the ordered mesoporous SBA-15 materials using the common synthesis method without the introduction of anhydrous ethanol.

To underpin the proposed deduction, a mixture solution of acid-leaching desilication solution and anhydrous ethanol was used to measure the polymerization extent of silanol species after aging 1 h, in which the volume ratio of two liquids was same as that of the synthetic solution for Sample S-E. As shown in Figure 7, an average size of 3.0 nm was detected for the silicon species, which was much smaller than that of silicon species existing in the acid-leaching desilication solution after the same aging time. Such a result sufficiently confirmed that the introduction of anhydrous ethanol with weak polarity could effectively inhibit the polymerization of silanol species. As a result, the silicon species existing in the synthetic solution possessed enough small sizes and rich surface hydroxyl contents. In addition, TG analysis was also used to compare the retained amount of P123 in different as-synthesized samples. As shown in Figure 8, one mass loss step was observed within the temperature range of 130–250 °C for the as-synthesized Samples S-E and S-H, which could be assigned to the decomposition of surfactant P123 that was retained within the mesopores [36]. Interestingly, compared to the as-synthesized Sample S-H, the as-synthesized Sample S-E exhibited a much steeper decomposition behavior of P123, which could be ascribed to the highly uniform dispersion of surfactant P123 in confined mesopores with an ordered arrangement (Figure 3a and Figure 5b). It was also noticeable that the as-synthesized Sample S-E showed a higher mass loss and higher initial decomposition temperature of P123 than those of the as-synthesized Sample S-H, indicating that the increased surface hydroxyl content of silicon species (Figure 6b) could significantly enhance the interaction between silicon species with surfactant P123 due to the introduction of anhydrous ethanol in the synthesis solution. Consequently, a larger amount of surfactant P123 in the synthesis solution acted as a template for the formation of ordered mesostructures. 

It is reasonably accepted that the abundant silanol groups in mesoporous walls of SBA-15 were beneficial to the grafting reactions between Si-OH and Al-OH for the preparation of Al-containing SBA-15 material with strong acidity [37]. In this regard, Sample S-E was used as a silica matrix for the incorporation of Al using the pH-adjusting hydrothermal treatment method [22], resulting in the successful synthesis of ordered mesoporous Al-SBA-15 material. As shown in Figure 3a, the resultant Sample SA-E exhibited three well-resolved diffraction peaks indexed as the (100), (110), and (200) reflections, indicating that the highly ordered mesostructure of Sample S-E was well-preserved during the hydrothermal grafting Al treatment process. A unit cell parameter (*a*) of 10.42 nm was calculated for the Sample SA-E (Table 2). The ordered mesostructure of SA-E was further reflected in the TEM images. The ordered 2D hexagonal arrangement of mesopores and the alignment of cylindrical mesopores were clearly observed along the [100] and [110] directions, respectively, as shown in Figure 4c,d. Sample SA-E displayed a typical Type IV isotherm along with a H1 hysteresis loop (Figure 5a) and a narrow mesopore size distribution (Figure 5b). As shown by the results presented in Table 2, the specific surface area, pore volume, and mesoporous size of SA-E were 871 m^2^/g, 1.14 cm^3^/g, and 8.34 nm, respectively. Further framework shrinkage of Sample S-E during the second hydrothermal treatment can be responsible for the decrease in the unit cell parameter and specific surface area for Sample SA-E, as compared with those of Sample S-E. In addition, compared to the matrix Sample S-E, the condensation between Si-OH and Al-OH and the formation of Si-O-Al bonding during the post-synthesis grafting Al process contribute to a remarkable decrease in the silanol concentration of Sample SA-E, reflected from its decreased integral area of the *ν*O-H band (Figure 6b), which can be responsible for the lower mass loss for Sample SA-E (Figure 8). It is worthy of noting that no characteristic reflection peaks associated with crystalline alumina were detected in the wide-angle XRD pattern of Sample SA-E (Figure 3b), suggesting that aluminum species had been highly homogenously incorporated into the amorphous mesoporous walls of Sample SA-E (Figure 4d). Such expected distribution of Al and Si instead of the presence of alumina aggregations was further confirmed by the element mapping images of Sample SA-E. As shown in Figure 9, for a randomly scanned region, elements O, Si, and Al were highly uniformly distributed throughout the whole sample. Compared with the initial Si/Al ratio of 20 for the synthesis of Sample SA-E, a final Si/Al ratio of 29 was measured by the inductively coupled plasma (ICP) analysis, indicating that most of the Al species existing in the synthesis solution could be, ultimately, homogeneously incorporated into the mesoporous silica walls of S-E.

Furthermore, NH_3_-TPD measurement was used to evaluate the surface acidity of SA-E, and Samples S-E and S-H were used for comparison. It can be seen from Figure 10a that different from Samples S-E and S-H that demonstrated an inconspicuous desorption peak of NH_3_, Sample SA-E exhibited a broad and asymmetric NH_3_ desorption peak within the temperature range of 120–400 °C, which corresponded to a wide distribution of acid strength. In terms of the difference in the desorption temperature of NH_3_, the acidic sites could be classified into three categories regarded as weak (100–250 °C), medium (250–350 °C), and strong (>350 °C) acid sites [22]. The Gaussian peak fitting method was used to deconvolute the NH_3_-TPD profile of Sample SA-E for comparing the distribution of acidic sites. As shown in Figure 10a, three fitted desorption peaks with the maximum temperature at 205, 275, and 405 °C were observed, suggesting that the introduction of Al atoms favored the generation of acid sites with different strengths. Moreover, the calculated fitted peak areas were 3.29, 4.02, and 1.02 (a.u.), respectively, indicating that the amount of desorbed NH_3_ mainly came from the acid sites with weak and medium acid strengths.

To further ascertain the types of acid sites on the surface of Sample SA-E, pyridine adsorption spectroscopy measurement was performed. As shown in Figure 10b, after evacuation at 150 °C, three sharp adsorption bands arising from the C-C stretching vibrations of pyridine were clearly discerned within the wavenumber range of 1600–1400 cm^−1^. It is widely accepted that the bands at 1547 and 1456 cm^−1^ can be attributed to the adsorption of pyridine molecules by surface B acid sites and L acid sites, respectively. Moreover, the band at 1492 cm^−1^ can be ascribed to the simultaneous adsorption of pyridine molecules by both surface B and L acid sites [22]. Evidently, Sample SA-E possessed both the B and L acid sites. The generation of B acid sites reveals that Al atoms had been successfully incorporated into the mesoporous silica framework of S-E to form bridging hydroxyl groups (Si-OH-Al; B acid sites) during the hydrothermal treatment process under weakly acidic conditions. Based upon the calculations, the total acid content was determined to be 0.871 mmol/g for Sample SA-E, in which the ratio of B acid sites to L acid sites was 0.33. The numbers of B and L acid sites with medium and strong acid strengths were further quantitatively estimated by degassing at 250 and 350 °C, respectively. From Figure 10b, it can be seen that the intensity of characteristic adsorption bands identifying B and L acid sites distinctly decreased with the increase in the evacuation temperature. This variation is more noticeable for the absorption band at 1547 cm^−1^, suggesting that the surface of Sample SA-E was rich in weak and medium L acid sites. After degassing at 250 °C, the calculated acid content with medium and strong strengths was 0.546 mmol/g, in which the quantity of the L acid sites was 0.486 mol/L. Further degassing at 350 °C, the numbers of strong B acid sites and L acid sites were calculated to be 0.016 and 0.126 mmol/g, respectively.

### 3.3. Removal of Methylene Blue Using the Prepared Mesoporous Materials

Methylene blue (MB) is a common toxic organic dye, and its massive discharge seriously endangers the aquatic ecosystem and exerts a drastic threat to human health [4]. The potential of Samples S-H, S-E, and SA-E derived from CFA in the removal of MB dye from simulated wastewater was evaluated using adsorption experiments and compared with that of ordered mesoporous material SBA-15 (Sample S-O). For an adsorption process, contact time exerts considerable influence on the adsorption kinetics, and the shorter equilibrium time corresponds to the higher adsorption efficiency. In addition, the pH value is a vital parameter influencing the adsorption efficiency of adsorbents. Thus, to investigate the effects of contact time and pH on the MB adsorption, 15 mg of absorbent (including S-O, S-H, S-E, and SA-E) was added to 50 mL of 50 mg/L MB solution with different pH values at 25 °C. As shown in Figure 11, for four samples at different pH values, the amount of MB adsorbed increased with the duration of the contact time, and the process of MB adsorption seemed to have occurred in two steps, namely, the rapid initial kinetic adsorption process occurring within 10 min, followed by an assuasive adsorption process. After 40 min of contact time, no evident increase in the adsorbed amount of MB was observed over the four absorbents. The high specific surface area and large mesoporous size (Table 2) endowed mesoporous silica materials (Samples S-O, S-H, and S-E) with a large amount of accessible active adsorption sites and a fine performance for the mass transfer of MB, which can be responsible for their high removal efficiency of MB during the initial adsorption process. As the adsorption sites were rapidly occupied by MB molecules, the adsorption capacity tended to stabilize. 

In addition, it can be found from Figure 11 that under different pH value conditions, the Samples S-H and S-E (especially Sample S-E) exhibited a much faster MB removal rate, as compared to Sample S-O. Among all mesoporous silica adsorbents, Sample S-E demonstrated the highest MB removal efficiency. The significantly improved MB removal performance for S-E can be attributed to its higher specific surface area than those of Samples S-O and S-H (Table 2), which exerted considerable influences on increasing surface active adsorption sites. Interestingly, the initial pH value of MB solution significantly affects the removal performance of mesoporous silica adsorbents. It was observed that with the increase in the pH of MB solution, both the removal rate and the adsorption capacity distinctly increased for Sample S-O, S-H, and S-E. As reported by the previous literatures [38,39], for the silica adsorbents, the surface Si-OH groups were identified as the active sites for the adsorption of cationic MB dye by the hydrogen bond and electrostatic interactions. When the pH is higher than the isoelectric point of silicon (2.5–3.0), the surface Si-OH groups of silica absorbents easily dissociate to surface Si-O^‾^ groups with a negative surface charge. With the increase in the pH of MB solution, the surface charge of Si-O^‾^ groups become more negative, which is more favorable for enhancing the electrostatic attractions between the silica adsorbent with the cationic MB molecules [4]. When a pH of 9.6 was used, a maximum MB removal efficiency of 90.5% and an adsorption capacity of 160.8 mg/g were observed for the Sample S-E. 

It is worth noticing that the highly homogeneous incorporation of Al onto the surface of Sample S-E provided a further boost in the removal performance of MB. The highest MB removal efficiency of 98.8% along with a MB adsorption capacity of 188.1 mg/g was obtained for Sample SA-E. Taking into consideration that Sample SA-E exhibited a relatively lower specific surface area than that of Sample S-E (Table 2), it is reasonable to believe that the introduction of Al atoms onto the surface of ordered mesoporous silica material played a positive role in the adsorption of MB. The presence of a large number of surface acid sites (Figure 10) from the doping of Al atoms can enhance the adsorption performance of MB through the coordination interaction between surface acid sites and MB molecules [40]. As a result, Sample SA-E possessed an optimal MB removal performance.

To determine the optimized amount of adsorbent, the dosage of Sample SA-E was varied in a range from 0.1 g/L to 0.5 g/L for the MB adsorption, at a pH of 9.6 and an initial MB concentration of 50 mg/L. As shown in Figure 12a, with the increase in the dosage of Sample SA-E, the removal rate and removal efficiency of MB increased remarkably, due to the increase in the number of active sites of adsorbent. When the dosage increases from 0.1 g/L to 0.3 g/L, the MB removal efficiency can increase from 39.0% to 92.8% within 5 min of contact time. Further increasing the dosage to 0.5 g/L, the adsorption equilibrium can be reached within 5 min, and a MB removal efficiency of 99.6% can be obtained, even though the MB adsorption capacity has decreased from 188.1 mg/g to 116.8 mg/g (Figure 12b). From these observations, an optimized dosage of 0.3 g/L can be determined.

The initial concentration of MB is another important parameter affecting the adsorption process, which also can be used for the determination of maximum adsorption capacity of adsorbents. The effect of MB initial concentration on adsorption capacity was studied by varying the MB concentration in a range of 10–70 mg/L at a pH of 9.6 and an optimized dosage of Sample SA-E. From Figure 13a, it can be seen that the removal efficiency decreases gradually from 99.9% for 10 mg/L to 86.7% for 70 mg/L with the increase in the MB concentration, during which the adsorption capacity increases gradually (Figure 13b). According to the effects of MB initial concentration on the removal efficiency and adsorption capacity, 50 mg/L was found to be the optimized concentration, and the maximum adsorption capacity of Sample SA-E was 190.1 mg/g.

## 4. Conclusions

Ordered mesoporous SBA-15 and Al-SBA-15 materials have been successfully synthesized using the silicon species derived from industrial waste coal fly ash (CFA) and were used as adsorbents for the efficient removal of organic dye methylene blue (MB) from an aqueous solution. The resultant SBA-15 material (Sample S-E) exhibited a highly ordered 2D mesoporous structure with a high purity, high specific surface area (1014 m^2^/g) and pore volume (1.08 cm^3^/g) and, therefore, displayed an excellent removal efficiency (90.5%) and adsorption capacity (160.8 mg/g) for methylene blue (MB) from simulated wastewater. In addition, the generation of surface acid sites from the homogenous incorporation of Al atoms into the mesoporous silica framework combined with the perfect retention of ordered mesostructures endowed the obtained Al-SBA-15 material (Sample SA-E) with a further boost in MB removal performance. For Sample SA-E, the MB removal efficiency reached almost 100% along with a maximum adsorption capacity of 190.1 mg/g, with that too within a contact time of only 40 min. The results obtained in the current work provides a new contribution in designing and utilizing CFA for the large-scale synthesis of ordered mesoporous SBA-15 type materials with excellent properties, which can be widely applied as effective and low-cost materials for adsorption and catalysis.

## Figures and Tables

**Figure 1 materials-16-02868-f001:**
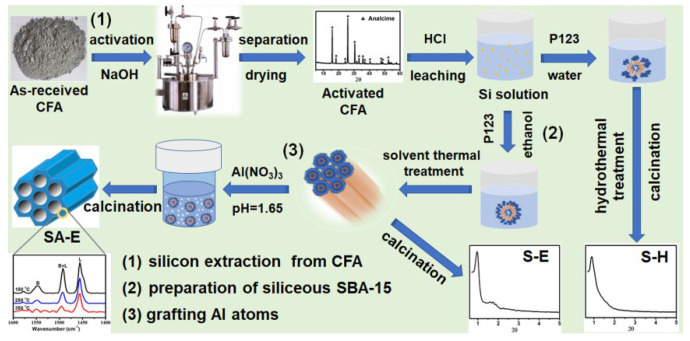
Schematic illustration for the synthesis of mesoporous materials of S-H, S-E, and SA-E.

**Figure 2 materials-16-02868-f002:**
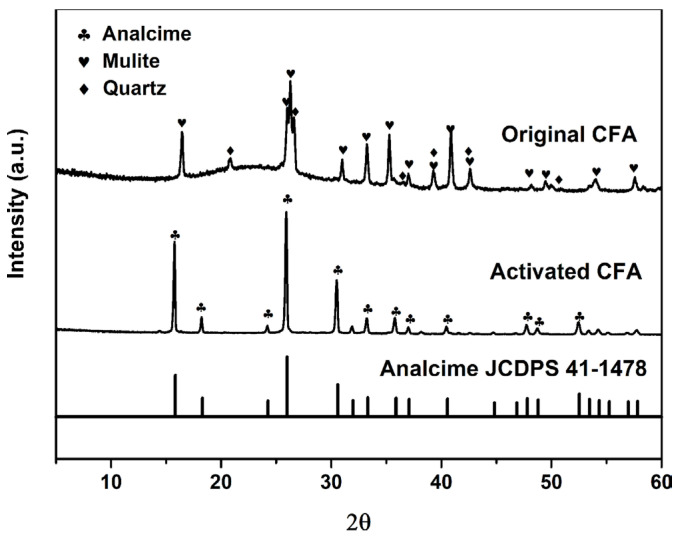
Wide-angle XRD patterns of CFA before and after the subcritical hydrothermal activation treatment.

**Figure 3 materials-16-02868-f003:**
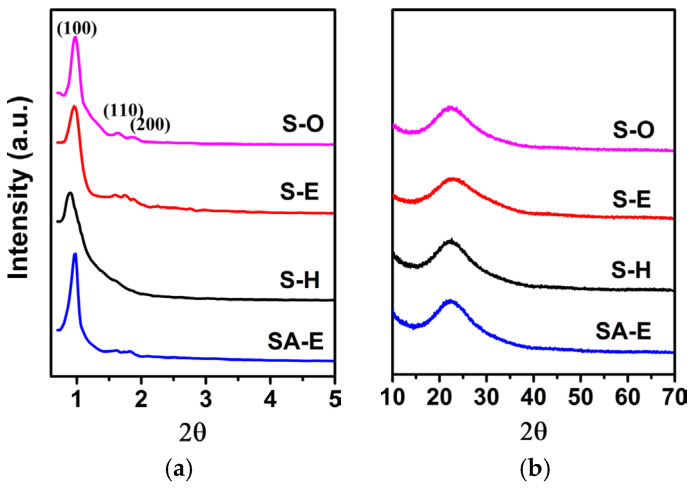
Small-angle (**a**) and wide-angle (**b**) XRD patterns of Samples S-O, S-H, S-E, and SA-E.

**Figure 4 materials-16-02868-f004:**
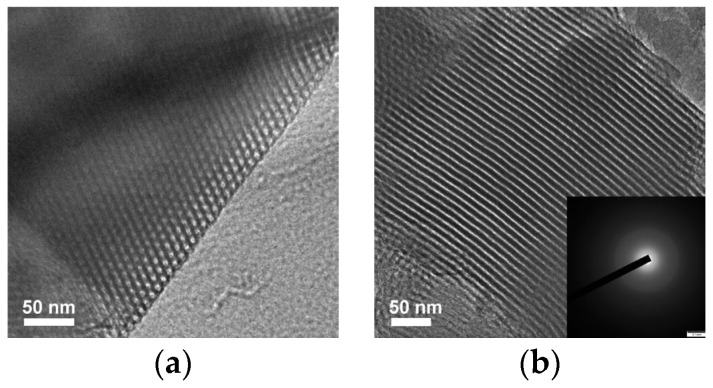
TEM images of Samples S-E (**a**,**b**) and SA-E (**c**,**d**) viewed along the [100] (**a**,**c**) and [110] (**b**,**d**) orientations, respectively.

**Figure 5 materials-16-02868-f005:**
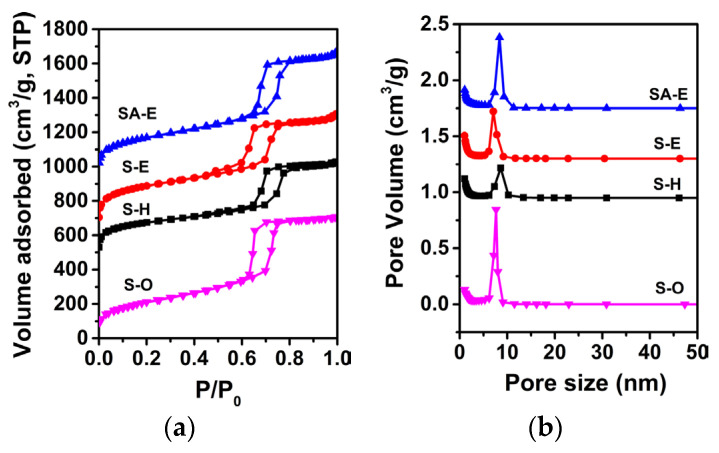
N_2_ adsorption–desorption isotherms and the corresponding pore size distribution curves of samples S-O, S-E, S-H, and SA-E. For clarity, in (**a**), the isotherms of S-H, S-E, and SA-E are offset along the Y axis by 440, 610, and 930 cm^3^/g, respectively. In (**b**), the Y axis values are 0.9, 1.3, and 1.7 cm^3^/g higher for Samples S-H, S-E, and SA-E, respectively.

**Figure 6 materials-16-02868-f006:**
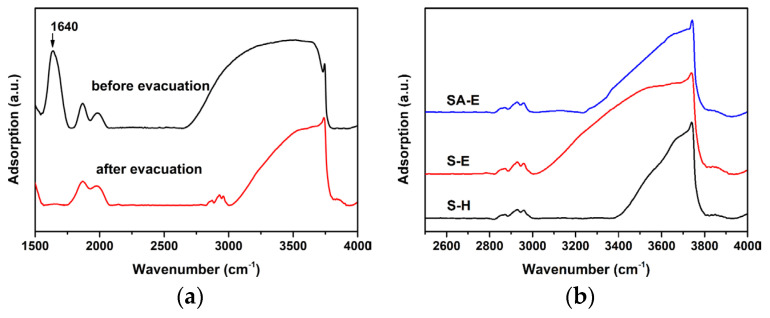
FTIR spectra of Sample S-E before and after the evacuation under vacuum (**a**); FTIR spectra of Samples S-H and S-E after evacuation under vacuum (**b**).

**Figure 7 materials-16-02868-f007:**
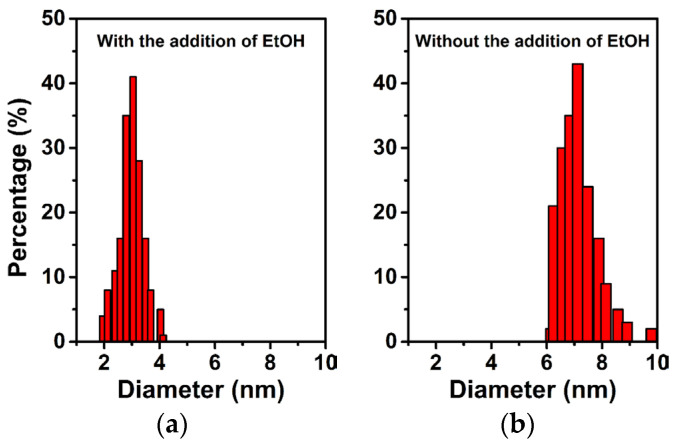
The size distribution of silicon species existing in the acid-leaching solution with (**a**) and without (**b**) the introduction of anhydrous ethanol.

**Figure 8 materials-16-02868-f008:**
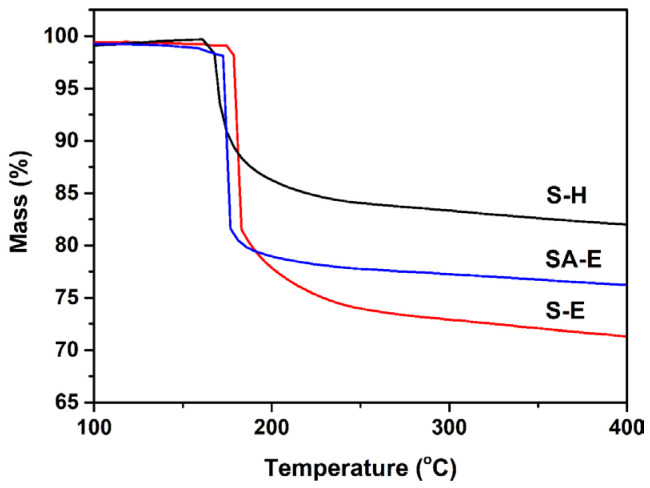
TG curves of as-synthesized Samples S-H, S-E, and SA-E.

**Figure 9 materials-16-02868-f009:**
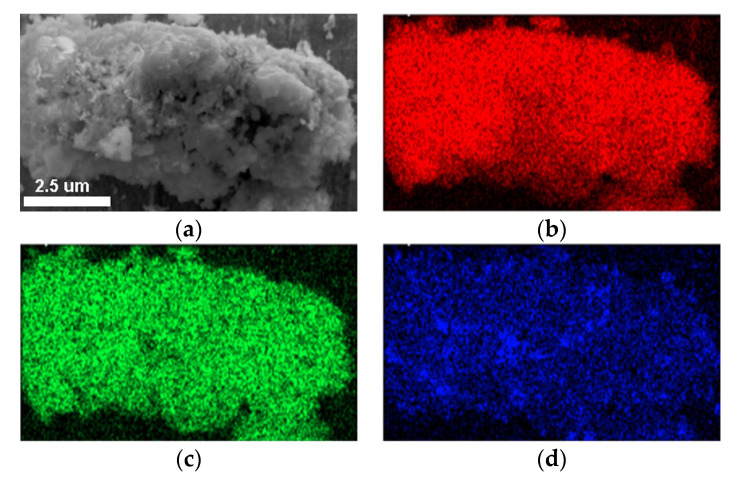
SEM image (**a**) and the elemental analysis mapping showing the distribution of O (**b**), Si (**c**), and Al (**d**) for the Sample SA-E.

**Figure 10 materials-16-02868-f010:**
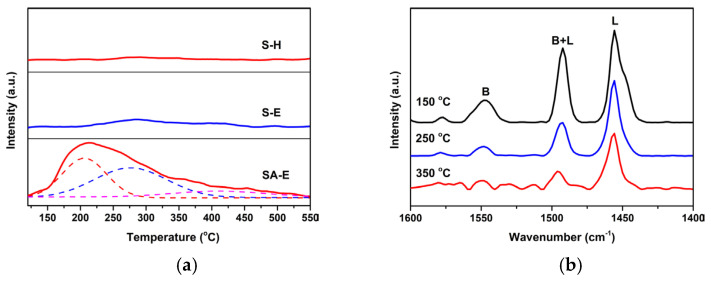
NH_3_-TPD profiles of Samples S-H, S-E, and SA-E (**a**), and pyridine-FTIR spectra of Sample SA-E after degassing at different temperature (**b**).

**Figure 11 materials-16-02868-f011:**
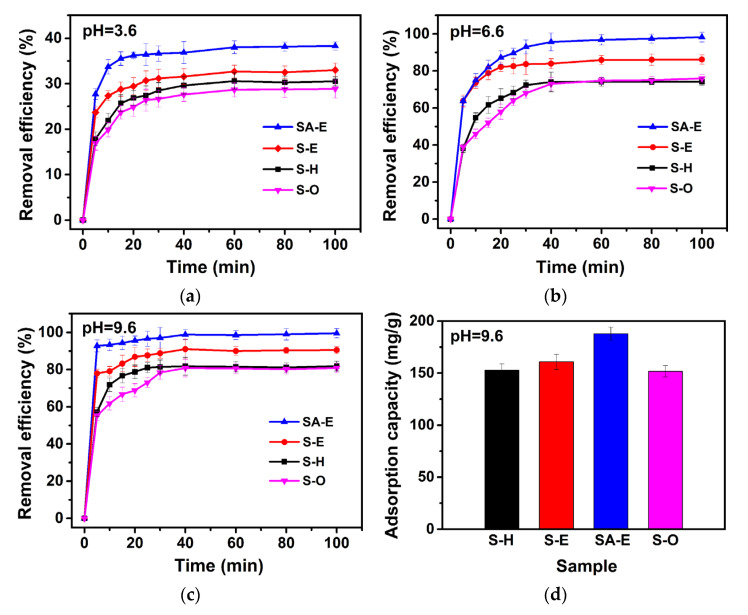
Effects of pH on the removal efficiency of methylene blue (**a**–**c**) and adsorption capacity of methylene blue (**d**) over Samples S-O, S-H, S-E, and SA-E.

**Figure 12 materials-16-02868-f012:**
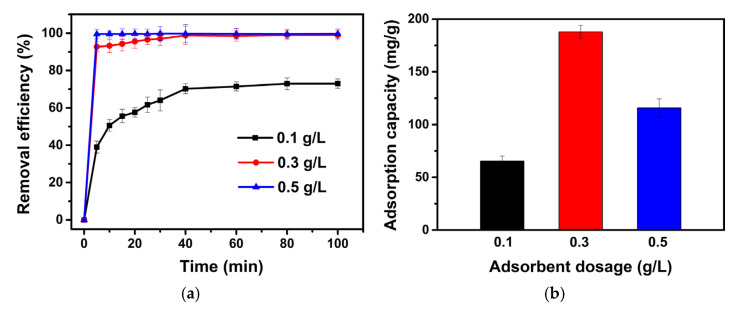
Effects of dosage of Sample SA-E on the removal efficiency (**a**) and adsorption capacity (**b**) of methylene blue.

**Figure 13 materials-16-02868-f013:**
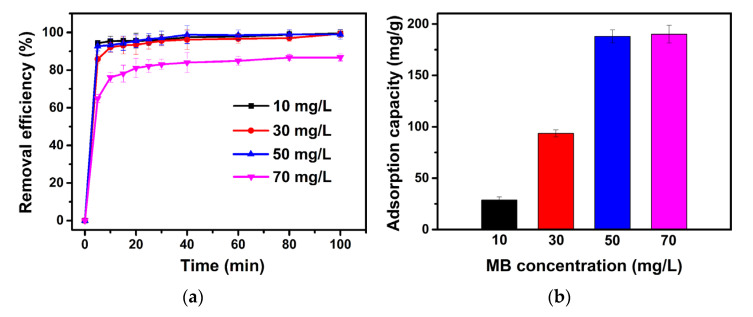
Effects of initial concentration on the removal efficiency (**a**) and adsorption capacity (**b**) of methylene blue over Sample SA-E.

**Table 1 materials-16-02868-t001:** Chemical compositions of coal fly ash before and after the activation (wt.%) and of silicon-rich solution (mg/mL).

	SiO_2_	Al_2_O_3_	Fe_2_O_3_	CaO	TiO_2_
Original CFA	47.9	36.3	6.2	3.1	1.7
Activated CFA	52.9	28.7	6.6	3.6	1.8
	Si	Al	Fe	Ca	Ti
Silicon-rich solution	7.22	8.26	1.03	0.95	0.43

**Table 2 materials-16-02868-t002:** Physicochemical properties of Samples S-H, S-E, and SA-E ^a^.

**Sample**	***d* (nm)**	***a* (nm)**	**S_BET_ (m^2^/g)**	***V*_p_ (cm^3^/g)**	***D*_p_ (nm)**
S-O	9.01	10.40	768	1.09	7.84
S-H	9.59	- ^b^	855	0.90	8.62
S-E	9.39	10.84	1014	1.08	7.06
SA-E	9.03	10.42	871	1.14	8.34

^a^: *d* represents the *d*-spacing calculated from the first strong diffraction peak; *a* represents the unit-cell parameter; *S*_BET_ represents the BET specific surface area; *V*_P_ represents the total pore volume; *D*_p_ represents the average mesoporous diameter. ^b^: The unit cell parameter of Sample S-H was not calculated due to its less mesostructural ordering.

## Data Availability

Not applicable.

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
