# Peer review of "Synthesis of Coal-Fly-Ash-Based Ordered Mesoporous Materials and Their Adsorption Application"

_materials, 2023, doi:10.3390/ma16072868_

Round 1
Reviewer 1 Report
The authors designed and synthesized ordered mesoporous SBA-15, derived from coal fly ash. The paper is clearly written. However, some of the terms and results need to be clarified.
Please explain and discuss what "green process" means.
Please also justify the lower energy statement in page 2, line 61 -63." our method can effectively increase the extraction rate of Si species combined with lower energy consumption and smaller usage amount of alkali.
Please also justify how this extraction is more efficient compare to the existing processes.
Section 2.4- the characterization procedures should be referenced
The TEM, XRD, N2 adsorption-desorption isotherms and the corresponding pore size distribution curves discussion lacks a comparison with the literature. Please compare your results
Figure 9 (a) and (b) include the error bar
Author Response
We thank Reviewer 1 very much for his/her positive comments. We have made revisions according to your comments . The point-by-point response please see the attachment.

Reviewer 2 Report
Journal: Materials (ISSN 1996-1944)
Manuscript ID: materials-2255820
Type: Article
Title: Green synthesis of coal fly ash-based ordered mesoporous materials with excellent properties and their adsorption application
Comments
The manuscript describes the synthesis of ordered mesoporous 14 SBA-15-type materials using coal fly ash (CFA) as raw material. Further Al impregnation has also been done on the same.
The manuscript is well written and shows successful formation of mesoporous silica which has been well supported by detail characterization. However similar synthesis, with the same hydrothermal technique and similar reagents have already been done.
However, the application part is an add on, but that requires a major revision.
1. The removal studies of MB has been shown wrt time variation. It should be done for other parameters as well, like adsorption dose, initial flouride conditions etc. You can refer to the following paper. Groundwater for Sustainable Development 14 (2021) 100622 and there are a no. of others with such studies
2. Title is too long and must be reframed.
3. Page 2, line 63. “Inspired from our previous work, herein, ordered mesoporous SBA-15, derived from CFA, was designed and successfully synthesized.
Give reference at a suitable place.
4. Page 2, line 85. “(2) preparation of as-synthesized siliceous SBA-15, and”
Should be written as “(2) preparation of siliceous SBA-15, and”
5. Page 6, line 220. “In addition, no diffraction peak was observed in the wide-angle XRD pattern of Sample S-E (Figure 3b), suggesting that the Sample S-E was amorphous in nature.” please check the statement and the sample nomenclature .
Author Response
We thank Reviewer 2 very much for his/her positive comments and helpful comment. We have made revisions according to your comments. The point-by-point response please seen the attachment.

Reviewer 3 Report
I would like to recommend this manuscript for publication in Materials Journal after the authors address several issues below
- The JCPDS numbers should be added to the XRD figures.
- Line 206-207 "Sample S-H prepared using water as the only solvent exhibited a strong peak at 0.90°in its XRD pattern, which implied a less mesostructural ordering." How does a stronger XRD peak indicate less mesostructural ordering?
- It is suggested to provide complete SEM images to clearly see the mesopore structure properties and determine their dimensions/shapes (0D, 1D, etc).
- The S-E sample has a higher BET surface area than SA-E, but its absorption capability is lower than SA-E. Why?
- How was the reusability performance of the samples? The authors should talk about it.
- The conclusion section is too lengthy. Please shorten
Author Response
We thank Reviewer 3 very much for your positive comments. We have made the revisions according to your comments. The point-by-point response please see the attachment.

Reviewer 4 Report
Green synthesis of coal fly ash-based ordered mesoporous materials with excellent properties and their adsorption application
The article discusses the synthesis of ordered mesoporous SBA-15 and Al-SBA-15, which were used as adsorbents to effectively remove the organic dye methylene blue (MB) from an aqueous solution. The silicon species used in the synthesis were derived from industrial waste coal fly ash (CFA). However, some recommendations should be taken into account for publication:
· The overall presentation quality is acceptable. The article is, on the whole, well-written; the English used in the paper is understandable.
Future recommendation:
1. To assess the environmental impact and pinpoint areas for improvement, a thorough life cycle assessment of the entire procedure, including the extraction of silicon species from CFA and the synthesis of SBA-15-type materials, should be carried out. The evaluation should consider energy use, carbon footprint, and potential pollutant emissions.
2. Optimizing the process parameters, such as the extraction conditions, synthesis conditions, and the co-solvent ratio, is crucial to ensuring that the process is scalable and reproducible. To ensure the process produces consistent results, a pilot study should be carried out to determine whether scaling up the procedure is feasible.
3. Toxicology tests should be carried out to assess the adsorbent's effects on the environment and human health to ensure its safety. Tests to determine the possibility of bioaccumulation and biomagnification should also be included, in addition to tests to determine acute and chronic toxicity. The risks should be reduced or eliminated if any toxicity problems are found.
Overall Merit:
The article certainly has some merit. For the rest, I believe that the article is organized in a logical and understandable manner.
Author Response
We thank Reviewer 4 very much for your positive comments. We have made the revisions according to your comments. The point-by-point response please see the attachment.

Reviewer 5 Report
The manuscript highlights new types of ordered mesoporous SBA-15 kind of materials derived from coal fly ash. The characterization and applicability of these materials for dye-polluted wastewater have been demonstrated. The manuscript has been presented well with necessary research findings. Yet, there are certain issues that need to be addressed, and a few more research studies are required for the article. Following are the comments for which the manuscript must be improved before it can be considered for publication.
· In general, green synthesis refers to the use of plant or biomass extracts for the production of working materials. Why do the authors call their approach green since the adopted method does not involve any biomass component for the production of the adsorbent? Indeed, the study uses a Supercritical-based activation cum extraction approach which is either a non-conventional or advanced category. Explain.
· Introduction must also highlight the need for wastewater treatment and adsorption as a viable technique for the same. Latest works on adsorptive wastewater treatment, such as Valorization of date palm leaves for adsorptive remediation of 2, 4-dichlorophenoxyacetic acid herbicide polluted agricultural runoff, and Biosilica/Silk Fibroin/Polyurethane biocomposite for toxic heavy metals removal from aqueous streams, can be considered for the same.
· Section 2.2: The conditions adopted in this study for the Supercritical based activation is different from the one followed by the authors in their previous studies. Besides, no information on the pressure for the supercritical process is not provided (although Fig-1 confirms it as a supercritical process). Is it a hydrothermal activation rather than supercritical (as in a supercritical process, a pressure greater than the critical pressure is usually maintained)?
· Line 92: Specify the temperature for the acid-assisted extraction of the silicon-rich solution
· Line 120: Ensure consistency in the denotations for S-H, S-E, and SA-E throughout the manuscript. For example, in line 120, there is a difference in the SA-E notation
· Introduction & Section 2.3: Why was MB used as a probe molecule in this study? Is there any industrial significance related to MB-polluted wastewater? For reasons of this kind, it is essential to explain the importance of wastewater treatment and adsorption as an easy solution for wastewater treatment in the Introduction section.
· Line 186 & Table 1: It is also observable that the acid treatment promoted better extraction of the aluminum also (from the activated CFA in spite of their lowered concentration due to activation) in addition to silicon extraction. How do authors call this solution a silicon-rich solution when indeed, aluminum is the dominant fraction?
· Lines 210-213 & Figure 3: The discussions don’t match the XRD patterns shown in Fig. 3a. Corrections are required.
· Figure 4: SAED patterns are further required to support the results and conclusion of the structural and compositional studies.
· Figure 3 & Figure 4: Authors must make significant discussions on the structure and composition of SA-E, which is currently missing in the article. Besides, FT-IR & TG results for SA-E must also be presented and discussed.
· Section 3.2: Although authors uphold the superiority of ethanol as a co-solvent over water through various characterizations, it is imperative that they explain the exact difference in the reaction chemistry (in terms of phase interactions) between that of ethanol and water when exposed to the silicon-rich solution.
· Section 3.3: Concerning adsorption, pH is a very important factor. What was the operational pH for the adsorption study? It is suggested that authors include pH dependency studies for all the adsorbents and prove that the performance series is still maintained as SA-E>S-E>S-H for different pH situations.
· Conclusions: Conclusions are highly narrative and must be effectively summarized to highlight the interesting findings of the work.
Author Response
We thank Reviewer 5 very much for your positive comments. We have made the revisions according to your comments. The point-by-point response please see the attachment.

Reviewer 6 Report
The manuscript presents a method used for the synthesis of ordered mesoporous SBA-15 type materials, based on a technology previously proposed by the authors for extracting Si species. Aluminum atoms were successfully incorporated into the silica framework by adjusting the pH hydrothermal grafting method.
The English language is good.
The manuscript is clear, well-structured and well referenced.
The objectives of the study are clearly stated and they have been achieved.
The Introduction section is comprehensive and provides relevant information about the topic.
The Materials and Methods section describes in detail the raw materials, the sample preparations and the test methods.
The experimental results are presented mostly in figures and they are analyzed in depth.
The conclusions are supported by results.
Minor observation
A comparison between the performance of the materials that the authors synthesized and the performance of other SBA-15 type materials used to remove the organic dye methylene blue from the aqueous solution would highlight the results obtained.
Author Response
We thank Reviewer 6 very much for your positive comments. We have made the revisions according to your comments. The point-by-point response please see the attachment.

Round 2
Reviewer 2 Report
The authors have addressed the comments raised and therefore the manuscript can be accepted for publication.
Reviewer 5 Report
Authors have amended the manuscript significantly based on my comments. Manuscript can be accepted for publication.